# Energy and Environmental Assessment of Steam Management Optimization in an Ethylene Plant

**DOI:** 10.3390/ijerph182212267

**Published:** 2021-11-22

**Authors:** Miroslav Variny, Kristián Hanus, Marek Blahušiak, Patrik Furda, Peter Illés, Ján Janošovský

**Affiliations:** 1Department of Chemical and Biochemical Engineering, Faculty of Chemical and Food Technology, Slovak University of Technology, Radlinského 9, 812 37 Bratislava, Slovakia; patrik.furda@stuba.sk (P.F.); jan.janosovsky@stuba.sk (J.J.); 2SLOVNAFT, a.s., Vlčie Hrdlo 1, 824 12 Bratislava, Slovakia; kristian.hanus@slovnaft.sk (K.H.); peter.illes@slovnaft.sk (P.I.); 3GRUCON, s.r.o., Nezábudková 24, 821 02 Bratislava, Slovakia; marek.blahusiak.pdc@gmail.com

**Keywords:** greenhouse gases, steam cracker, energy management, steam let-down, fuel consumption, carbon tax

## Abstract

Steam crackers (ethylene plants) belong to the most complex industrial plants and offer significant potential for energy-saving translated into the reduction of greenhouse gas emissions. Steam export to or import from adjacent units or complexes can boost the associated financial benefit, but its energy and environmental impact are questionable. A study was carried out on a medium-capacity ethylene plant using field data to: 1. Estimate the energy savings potential achievable by optimizing internal steam management and optimizing steam export/import; 2. Quantify the associated change in air pollutant emissions; 3. Analyze the impact of the increasing carbon price on the measures adopted. Internal steam management optimization yielded steam let-down rate minimization and resulted in a 5% (87 TJ/year) reduction in steam cracker’s steam boiler fuel consumption and the associated cut of CO_2_ emissions by almost 4900 t/year and that of NO_x_ emissions by more than 5 t/year. Steam import to the ethylene plant from the refinery proved to be purely economic-driven, as it increased the net fuel consumption of the ethylene plant and the refinery complex by 12 TJ/year and resulted in an increase of net emissions of nearly all considered air pollutants (more than 7000 t/year of CO_2_, over 15 t/year of NO_x_, over 18 t/year of SO_x_) except for CO, where the net change was almost zero. The effect of external emissions change due to the associated backpressure electricity production surplus (over 11 GWh/year) was too low to compensate for this increase unless fossil fuel-based electricity production was considered. The increase of carbon price impact on the internal steam management optimization economics was favorable, while a switch to steam export from the ethylene plant, instead of steam import, might be feasible if the carbon price increased to over 100 €/tCO_2_.

## 1. Introduction

### 1.1. Overview and Literature Survey

The deepening climate crisis makes it imperative to substantially reduce anthropogenic carbon emissions. Reduced industrial emissions production has the potential to contribute visibly to achieving this goal, as they globally represent around a third of total anthropogenic emissions [1,2]. Around 40% of industrial emissions can be attributed to the production of energies and media consumed in production processes [3,4]. Electricity and steam production, distribution, and consumption systems are ideal candidates for design and operation optimization [5,6] within the scope of energy management systems implementation [7,8]. The drivers of such activities include several technical and non-technical aspects, such as sustainable industrial production [9,10,11], labeling of products as ‘green’ [12], increasing price of carbon emissions [13,14], as well as the persistent need for an increase in production efficiency [15] to remain innovative and competitive on the market [16,17] and to comply with the legislation [18].

In addition to electric energy, virtually consumed in any production process, the production of heat carriers contributes to both operational expenses and greenhouse gas (GHG) emissions [19,20]. Among these, steam has a prominent position especially in heavy industries, including but not limited to: steel and ironmaking [21,22], pulp and paper [23,24], ammonia and fertilizer production [25], refining and petrochemistry [26,27]. As it results from the historical development of both technologies and industrial plants, steam production in such industries is generally a mix of centralized and decentralized sources [28]. The central industrial combined heat and power plant (CHP) serves as a marginal steam source, while a substantial amount of steam is produced and consumed directly at the production plants [29]. Industrial steam networks are often complicated systems of interconnected pipelines with steam sources and consumers connected to several steam pressure levels [30,31]. Managing such systems requires suitable decision-making tools and sufficient flexibility, which is usually provided by CHP and process steam drives that can be alternated with electrodrives [32,33].

Considering the refining and petrochemical industry, individual production units are typically assembled in organized production groups, refineries [34], that can be a part of even larger chemical complexes and industrial parks [35,36]. Typically, three to five main steam pressure levels satisfy the needs for delivered steam quality [6]. CHPs are commonly equipped with several steam boilers producing high-pressure steam and several steam turbines to cope with the varying demand for steam at individual pressure levels [37]. Liquid fuels (heating oils, residues of conversion processes) or gaseous fuels (refinery gases, offgases, and natural gas) are fired in such boilers. Steam turbines cogenerate electric energy, covering a part of the electricity demand of the refinery, while extracted and backpressure steam is partly used in the CHP and the rest is exported to the refinery [38,39]. On the other hand, decentralized steam production in production units is rarely carried out on purpose; residual heat content of material streams or flue gas from refinery furnaces is an almost exclusive heat source for steam production [40]. Among these facilities, ethylene plants (Steam Crackers, SC) are examples of deeply energy- and process-integrated units, combining high-pressure steam production in convection sections of pyrolysis furnaces and dedicated steam boilers [33,41,42]. Produced steam is expanded in large steam turbines driving key process equipment [43]. The extracted steam is used in smaller process steam drives or for heating and other purposes [44,45]. The steam system in such plants is supplemented by let-down stations and steam export to or import from the main steam network is possible if the plant is integrated into a larger industrial complex [46,47]. Parallel production of steam at various pressure levels and mechanical energy [48,49] in SC can be, together with the production of steam and electric energy in industrial CHP, termed cogeneration while the whole system, which produces energies and materials, is a polygeneration plant [50,51]. The efficiency of heat and power production (mechanical energy), as well as fuel costs and GHG emissions in an industrial CHP and SC, can vary over time due to many factors. This opens the possibility of combined energetic-economic-environmental (3E) optimization [52,53]. Integration of petrochemical plants and refining plants comprises several material streams—feed [54], hydrogen [55], offgases [56], semi-finished products and products [46,57], and utilities [58], and thus poses an extraordinarily complex optimization problem [59].

Zhao et al. (2017) [46] and Ketabchi et al. (2019) [47] explored the potential of material integration of an ethylene plant in an oil refinery. Taking advantage of the synergies of both plants’ operations, a significant increase in their profit could be achieved based on optimization calculations. The most recent studies by Gong et al. (2019) [57] and Dai et al. (2021) [53] focused on material efficiency within an ethylene plant either by optimizing the reactor part [53] or the back-end separation part [57]. Shen et al. [58] proposed a framework for optimizing the energy structure of a standalone ethylene plant, comprising multiple steam pressure levels, steam sources, consumers, and steam drives operating between several pressure levels. Optimization of such a system led to increased use of electrodrives instead of steam drives and to a reduced amount of steam sent through let-down stations, whereby fuel energy consumption reduction of around 14% could be achieved. Zhao et al. [33] successfully applied similar proposals in a study on optimizing the operation of another ethylene plant. In a more recent study, Shen et al. (2021) [60] optimized a part of the SC gas plant in terms of exergy efficiency and processing costs. Wang et al. (2020) [61] proposed a framework for multiobjective optimization of production plants and applied it to the cryogenic separation part of an ethylene plant. As a result, a tradeoff was found between increased propylene production and reduced operational expenses. Numerous recent studies document the ongoing intense effort to reduce the operational expenses and to increase the material and energy efficiency of ethylene plants.

### 1.2. Contribution of This Study

The above survey has shown that optimizing energy consumption and process features in steam cracker plants is a promising means of achieving material and energy efficiency improvement goals. However, many studies dedicated to optimizing ethylene plants focus solely on the CO_2_ balance or omit the environmental problems entirely. As examples, studies by Li et al. (2013) [41], Gong et al. (2019) [57], Shen et al. (2019) [58], and Zhao et al. (2019) [33] delivered important findings related to the potential reduction in energy consumption as a result of optimized SC operation, while Chen et al. (2020) [62], Geng et al. (2020) [59], and Dai et al. (2021) [53] explored both energy efficiency and CO_2_ emissions aspects. This is in line with recent developments in industrial processes and energy systems optimization [63,64], where the calculation of CO_2_ emissions is a widely adopted means of assessing the environmental impact [52,65]; only rarely supplemented by calculation of other GHG emissions [66,67] or expressed via CO_2_ equivalent [68,69]. The analysis of GHGs other than CO_2_ is, on the other hand, common in the evaluation and optimization of combustion processes and power plants [70,71].

To the best of our knowledge, there are no relevant studies in the field of SC energy management optimization addressing other major pollutants and employing field data at the same time.

Thereby, a knowledge gap is identified, namely:Whether and how are energy savings achievable by internal SC energy management optimization affected by the possibility of steam import/export to/from an SC?What does the resulting balance of the main air pollutants look like, and to what extent can the feasibility of energy costs saving measures be affected by the increasing carbon tax?

The aim of this study is to provide answers to these questions with the help of field data from a real SC integrated in a refinery.

## 2. Materials and Methods

### 2.1. Steam Cracker Steam System

A mid-capacity Steam Cracker located in the European Union was selected as a model case. It is material- and energy-integrated in a refinery and processes both gases and gasoline. It has a complex steam system that can act as a standalone system or, if needed, is capable of importing or exporting steam from/to the outer steam network as depicted in Figure 1. The steam network operates at six different pressure levels with the following typical parameters.

SS (superhigh pressure) steam: 11.0 MPa (g), 520 °C;HS steam: (high pressure) 3.5 MPa (g), 350 °C;MS steam: (intermediate pressure) 1.2 MPa (g), 250 °C;LS steam: (low pressure) 0.4 MPa (g), 170 °C;PS steam: (very low pressure) 0.25 MPa (g), 138 °C;DS steam: (dilution steam) 0.7 MPa, 190 °C.

SS steam is produced internally in cracking furnaces and in an auxiliary steam boiler. The contributions of steam produced in the furnaces and boiler were almost equal in the past, as shown in Figure 2. 

However, the share of steam produced in furnaces increased in recent years, especially due to increased SC feedstock processing (2017–2019). The corresponding effect can be seen in Table 1 in the decreasing values of SS steam consumption per % of specific ethylene production, which indicates that the plant operates more efficiently at higher throughputs. SS steam is consumed in combined extraction-condensing steam drives driving process gas compressors. Excess steam is reduced in the let-down station to HS steam. Similarly, HS and MS steam is used to drive various steam drives, and the rest is reduced to lower pressure level steam. LS steam is used mostly as a heating medium in fractionators, boiler feed water (BFW) preparation, or steam tracing. In addition, HS, MS, and LS steam export or import are possible; however, only HS and LS steam import are usually used. PS steam is produced by LS steam reduction and is used as a heating medium in some heat exchangers and vessels. DS steam is used in the process as a dilution stream added directly to the feedstock.

The increasing share of steam throttling presented in Table 2 is an indication of a nonfunctional regulation element. Let-down valve ‘A’ lost its regulation ability due to erosion under extreme operating conditions. The valve opening was rather constant at 14% before the turnaround in 2019, yet the steam inlet flows increased significantly within one year.

An external energy audit performed in 2013 recommended the installation of a parallel let-down valve with a smaller capacity and an independent regulation to ensure smooth and functional regulation of the SS steam network. This was in line with the best practice, which recommended reducing the amount of steam flowing through let-down stations to a minimum. Preliminary analysis found two possible benefits:

Direct fuel (natural gas) savings are expected due to more efficient co-generative HS steam production by extraction from steam drives instead of SS to HS steam throttling, translated into a reduced condensing load of the steam drives. Thus, an improvement in the energy efficiency of SC and the whole refinery is expected. In addition, fewer GHG emissions will be produced.HS and LS steam import will be enabled due to less steam being produced internally. Imported steam is usually produced from cheaper fuel (mixed petroleum residue, MPR, which is usually in excess). Additional extraction electricity generation will be possible thanks to the increased steam supply from the CHP unit. These effects contribute to better economic results. On the other hand, there is a negative environmental effect of MPR fuel utilization, as its combustion produces more greenhouse gases (GHGs) than that of natural gas.

Implementation of the energy audit recommendations took almost six years.

### 2.2. Combined Heat and Power Unit

Combined heat and power unit (CHP) serves as a marginal steam source for the refinery, balancing steam demand and supplying steam at three pressure levels: 3.5 MPa (g) (HS), 1.0 MPa (g) (MS), and 0.4 MPa (g) (LS). It comprises several MPR-fired steam boilers that supply high-pressure steam (9 MPa, 535 °C) to a battery of steam turbines producing backpressure and condensing electric energy and steam at various pressure levels, both for own consumption and for export. LS export is the most sensitive to ambient temperature as it is predominantly used for space heating and steam tracing of process streams. During warmer months, especially from June to September, LS is occasionally vented into the atmosphere to prevent a stall in the LS steam mains and thus to avoid excessive steam condensation followed by corrosion and potential equipment failure. A more detailed description of the layout and operation of the CHP unit, including the management of the steam network in the refinery, can be found in our previous studies [28,72].

There is a possibility of supplying steam from the CHP unit to SC at all three main steam pressure levels, but an autonomous SC operation in terms of steam production and consumption was preferred in the past. Optimized SC energy management opens the possibility of importing steam from the CHP unit if it is economically attractive. Given the situation of LS excess in the refinery in the summer months, this would affect mainly the LS pressure level.

### 2.3. Energy and Environmental Assessment

#### 2.3.1. Steam Cracker

Rerouting SS steam from the let-down station to the steam turbine increases the production of backpressure energy and, thus, reduces the needed condensing load. Analysis of the steam turbine´s steam consumption diagram and the steam expansion path in the enthalpy-entropy diagram of steam, according to the procedure proposed by Medica-Viola et al. (2020) [73], revealed the following:Rerouting 4 t/h of SS steam from the let-down station to the steam turbine and letting it expand to the HS pressure level reduces steam flow to the turbine condenser by 1 t/h;Heat rejection in the steam turbine condenser is reduced by 2.268 GJ/h per 1 t/h of steam flow to the condenser, considering the enthalpy of the discharge steam to the condenser of 2.438 GJ/t and that of the steam condensate leaving the condenser of 0.168 GJ/t.

The following data express the effect of fuel savings and the reduction of GHG emissions released from the SC boiler:SS steam enthalpy leaving the SC steam boiler is 3.435 GJ/t and the difference between this value and that of steam condensate enthalpy leaving the turbine condenser has to be supplied by the fuel combusted in the boiler;Thermal efficiency of the boiler resulting from certified measurements is 94.1%, based on the lower heating value of the combusted fuel (natural gas, NG);The lower heating value of NG is 49 GJ/t and its emission factor is 2.75 tCO_2_/tNG.

Emissions of other GHG released from the SC boiler can be evaluated using the data in Table 3. Data were obtained from the automated monitoring system (AMS) of the SC.

#### 2.3.2. Combined Heat and Power Plant

CHP serves as a marginal steam source and thus any change in the SC energy management patterns leading to steam import or export affects the CHP operation. According to our previous studies [28,72], the following data were adopted.

Specific MPR consumption is 0.09 t per ton of exported steam;The lower heating value of MPR is 40.3 GJ/t and its emission factor is 3.2 tCO_2_/tMPR;The average steam enthalpy imported to the SC from the CHP is 2.9 GJ/t;Marginal backpressure electricity production in the CHP amounts to 150 kWh per ton of exported steam.

Emissions of other GHGs released from the CHP can be assessed using the data in Table 4 that were obtained from the refinery AMS.

The change in the backpressure electricity production in the CHP is associated with the change in the GHG emissions from electricity production elsewhere. To quantify this effect, the average energy mix of Slovenské Elektrárne, a.s., (Bratislava, Slovakia), the main electricity producer in Slovakia, was applied. The corresponding emission factors are presented in Table 5.

## 3. Results and Discussion

### 3.1. Energy Consumption

Table 6 provides key figures related to the change in fuel consumption in the SC after the installation of a smaller SS to HS let-down valve in 2019. As it results from Table 6, ethylene production remained almost constant during 2020 but the amount of SS steam let-down to HS decreased visibly. The first quarter of 2020 was affected by SC staff who were still learning how to fully exploit the potential offered by the smaller valve ‘B’. From the second quarter of 2020 onwards, the amount of SS steam let-down to HS was reduced by more than 75% compared to the average value shown in Table 2. The larger let-down valve ‘A’ is closed most of the time and the opening of valve ‘B’ is around or below 50%, indicating a functional SS steam let-down regulation. The associated decrease in NG consumption reached around 9000 GJ (180 t) in the first quarter of 2020 but increased to almost 29,000 GJ (580 t) in the next periods. The total estimated NG savings due to the decrease in the SS to HS steam let-down in 2020 compared to 2018 amounted to around 87 TJ (1778 t).

In addition to lower SS to HS steam let-down rate, the SC implemented changes in the energy management aimed at optimizing the fuel costs; i.e., increased import of steam from the refinery to reduce NG consumption in the SC boiler. The resulting impact on fuel consumption in both SC´s steam boiler and CHP´s steam boilers is shown in Figure 3. A comparison of NG energy consumed in the SC boiler in 2018 and 2020 yielded a decrease of more than 470 TJ. Its decomposition into individual contributions showed that, as presented in Table 6, 87 TJ are attributable to the decreased SS to HS let-down, which represents around 18% of the total change in NG consumption and an about 5% decrease in the total fuel consumption in the SC steam boiler at the same time. Additionally, SC steam import increased by 225 TJ in 2020 compared to 2018, which resulted in another decrease in NG consumption in the SC boiler by 270 TJ. This represents the largest contribution to the total NG consumption achieved. The remaining NG consumption decrease of 116 TJ (25% share in the total change in the NG consumption) was attributed to other investments and operational optimization activities performed during the SC turnaround in 2019. Ethylene production in 2018 and 2020 was almost the same, so it did not affect the changes observed in fuel consumption.

The increase in steam import from the refinery to the SC in 2020 compared to 2018 caused changes in the operation and load of the CHP (Figure 3). To cover the change in steam demand and condensing power production requirement, annual fuel energy consumption in CHP increased by more than 400 TJ between 2018 and 2020. Of this, 282 TJ (almost 70%) can be attributed to increased steam import to SC and the remaining 129 TJ (around 30%) represent other factors. It should be noted that the relative change in fuel energy consumption in the CHP is relatively small (around 4%) and the one attributable to the change in SC energy management is even smaller (less than 3%). Therefore, the effect of the change in SC steam import could be recognized in the CHP fuel balance only because of analogous steam consumption patterns in the refinery and comparable condensing power production in the CHP in 2018 and 2020.

Another interesting fact resulting from the data presented in Figure 3 is that the increase in steam import to SC did not result in a reduction in net fuel consumption. NG consumption in the SC steam boiler decreased by 270 TJ while that in the CHP steam boilers increased by 282 GJ. Thus, the import of steam from the refinery to the SC is a purely economic issue, which results from the 2020 NG price per GJ energy being by 20–40% higher than that of MPR fired in the CHP boilers. With a different fuel price constellation, the SC might start exporting steam to the refinery. In contrast, the installation of a smaller SS to HS let-down valves yielded net fuel energy savings localized in the SC. Therefore, this energy measure is feasible both from the economic and the energy point of view regardless of fuel cost. Moreover, there is neither negative nor positive synergy between these two measures, so that they can be exploited in parallel with a clearly distinguishable economic and energy effect. 

### 3.2. GHG Emissions

The change in SC and CHP operation patterns resulting from the implemented SC energy management measures caused a change in GHG emissions. Figure 4 presents the resulting CO_2_ balance divided into individual items. The total annual change in CO_2_ emissions from the SC steam boiler resulting from the decrease in annual fuel energy consumption of 473 TJ (Figure 3) amounted to more than 26,500 t. Of this, almost 4900 t can be attributed to the decrease in SS to HS steam let-down and, since there is no change in CO_2_ emissions at the CHP, it represents the net CO_2_ balance of this measure for the refinery and SC complex at the same time. On the contrary, the increase in steam import to the SC resulted in a net increase in CO_2_ emissions for the refinery and SC complex of more than 7000 t, as the decrease in CO_2_ emissions of more than 15,000 t achieved in the SC was outbalanced by their increase by more than 22,000 t at the CHP. This finding further underlines that, while the achieved decrease of SS to HS steam let-down is beneficial from all considered viewpoints (economics, energy, environment), the increased steam import to the SC is only a matter of economics while its energetic end environmental impact is negative.

However, it must be remembered that the increase in CHP steam export caused by the increase in SC steam demand resulted in an increase in backpressure electricity production in the CHP which amounted to more than 11,000 MWh/year. Accepting the assumption that the refinery’s electricity consumption was not affected, an equivalent decrease in the purchase of electricity from the outside grid was achieved. This further led to a decrease in electricity production elsewhere. Applying the CO_2_ emission factor of the energy mix of Slovenské Elektrárne, a.s. (Bratislava, Slovakia), (Table 5), this translated into an almost 1600 t decrease in CO_2_ emissions outside of the refinery and SC complex. Thus, extending the control volume for the CO_2_ balance calculation to the entire Slovak republic yielded the resulting final change in CO_2_ emissions of over 5600 t/year. It can be concluded that even this CO_2_ balance disfavors the steam import to the SC from the refinery. An alternative approach, using marginal emission factors, is presented in Section 3.4.

Considering the CO_2_ balance effects of the SS steam let-down minimization and the increased steam import to the SC, a value below 1 kt/year was obtained (see Figure 4). This is a favorable result since the resulting combined effect of both actions is almost CO_2_ neutral while being economically attractive and leading to net fuel energy savings.

The balance of other major GHGs was also estimated using the same approach and the results are presented in Table 7. Given the low values of CO and SO_x_ emission factors from SC steam boilers resulting from clean fuel combustion (NG) [74], the only notable reduction in emissions from SC is that of NO_x_. This amounts to more than 5 t/year in case of decreased SS to HS let-down and more than 15 t/year in case of steam import to the SC. On the other hand, there is a significant increase in SO_x_ and NO_x_ emissions from the CHP steam boilers, which results from an increase in the CHP load due to the import of steam to the SC amounting to more than 30 t/year regarding NO_x_ emissions and more than 18 t/year regarding SO_x_ emissions, respectively. This is caused by MPR fired in the CHP steam boilers being not clean fuel. The resulting emissions of sulfur oxides and nitrogen oxides are higher than those from NG combustion despite the renovation of the CHP in the past coupled with the commissioning of a flue gas desulfurization unit [75]. Even if the benefit of backpressure electricity production is incorporated in the calculations, emissions of both SO_x_ and NO_x_ increased by more than 13 t/year each; only those of CO decreased slightly. It can be concluded that while the reduction of SS to HS steam let-down led to a decrease in all considered GHGs, increased steam import to the SC led to their increase, except for the case of CO.

The given refinery is dedicated to transparency and committed to reducing the environmental impact of its operation. Like several other refineries and petrochemical enterprises, it publicly provides data on the GHGs and other emissions [76] and has implemented several investment projects to increase the energy efficiency and reduce the carbon footprint of both the SC [77] and the CHP [75] units. As a part of a business group, it actively participates in the long-term strategy of the company [78] to become a net-zero CO_2_ emitter. Energy management of the refinery and SC complex is expected to become a part of a larger optimization scheme that includes the entire business group, supporting the long-term goals of the strategy.

### 3.3. Economic Considerations

Though exact values are confidential, a rough economic evaluation of both measures together yields an annual cash flow of around 1 mil. € with approximately 70% of it resulting from decreased SS to HS let-down. This value already incorporates the CO_2_ effect. Considering the CO_2_ balance of both measures, an increase of CO_2_ price by 50 €/t (i.e., roughly doubling its summer 2021 price) would increase the cash flow of the decreased SS to HS let-down by almost 250,000 €/year, while that of the increased steam import to the SC would drop by over 350,000 €. Under such conditions, steam import to the SC is no longer beneficial unless the price difference between NG and MPR further increases. Though the observed and continuing trend of CO_2_ price increase is remarkable, its impact on the economic results is limited. A much higher effect on the project economics has the currently (October–November 2021) extremely high natural gas price which leads to the maximization of the steam import to the SC, despite the associated net increase in CO_2_ emissions.

The price difference between the NG and MPR price (in €/GJ) evolves dynamically; it increases nowadays as a result of the currently increasing NG prices worldwide. Despite the favorable actual situation, an opposite one has occurred several times in the past when the NG price dropped below that of the MPR. In such a case, the SC can export steam to the refinery at the HS pressure level. The technical issues associated with the operation of the HS steam network in the refinery were described in our previous study [28]. Additional steam from sources other than the CHP can be absorbed only occasionally and surplus HS steam let down to the MS pressure level. The effect of letting steam down from the HS to the MS pressure level has to be accounted for and it lowers the resulting benefit.

### 3.4. Discussion

As it results from Figure 3 and Figure 4, increasing the import to the SC from the refinery at the expense of own steam production is purely an economic driven measure. From a technical point of view, it can be considered an inter-fuel substitution [79]. The energy and environmental aspects of fuel substitution were studied by several researchers. Ditaranto et al. (2013) [80] explored the substitution of natural gas by hydrogen in fired heaters in a refinery using CFD simulation and experimental setup. They found that nitrogen oxides emission can be substantially reduced even if the whole hydrogen production chain was considered. Glushkov et al. (2020) [81] analyzed the switch of fuel from coal to composite fuel in a power plant to reduce greenhouse gas emissions, experimentally confirming significantly lower SO_x_ and NO_x_ emissions in the case of composite fuel combustion. Jou et al. (2010) [82] studied the possibility of recovering tail gas from several refining units and its use as a partial replacement of NG in refinery heaters, finding that, contrary to the significant reduction in CO_2_ emissions, the level of NO_x_ increased modestly. Although the research presented suggests a transition to cleaner fuels as a viable means of reducing GHG emissions, a transition in the opposite way might still be acceptable for the industry if it is economically justified [38]. Moreover, as presented in Figure 4, there is a possible associated effect of the change in electricity production if fuel substitution is performed between two distinctive utility systems. Although both the SC and the CHP operate in the cogeneration mode, the SC produces mechanical power, while the CHP produces electricity. In such a case, the external emissions change should also be considered, as presented in Figure 4 and Table 7. 

Emission factors from electricity production are known to vary significantly depending on the energy mix and other conditions considered in their evaluation [83,84]. As discussed in our recent study [85], the emission factor of CO_2_ can vary between close to zero values for an almost exclusively renewable energy-based mix up to 700 to 900 kg/MWh for coal- and oil-fired power plants. Adopting a value of 700 kg/MWh [86] instead of that from Table 5 (136 kg/MWh) substantially changes the resulting global CO_2_ balance of increased steam import into SC: global increase in CO_2_ emissions of 5600 t/year changes to the reduction of approximately 900 t/year. Similarly, much higher CO, NO_x,_ and SO_x_ emission factors were reported than those considered in this study (Table 5) for fossil fuel-fired power plants [87,88]. Their application in the GHG balance related to increased steam import to SC presented in Table 7 would lead to a net reduction of the emissions of all GHGs considered, although a net increase in both SO_x_ and NO_x_ emissions was observed for the refinery and SC control volume.

Calculation of the emissions of major GHGs and assessment of their reduction in the design phase or by revamp, renovation, and operation optimization of existing plants is a part of the environmental assessment of several studies. Often, emissions of all major pollutants considered are expressed in form of CO_2_ equivalent. Bamufleh et al. (2012) [68] and Luo et al. (2014) [89] applied this concept to assess the optimal layout of a cogeneration unit by means of multiobjective optimization. Alhajji and Demirel (2015) [69], Comodi et al. (2016) [90], and Berghout et al. (2019) [29] attempted to design and operation optimization of various refinery units and utility systems and used CO_2_ equivalent calculations as one of the decision criteria in the optimization process, considering either model emission factors provided by modeling software [69], extracted them from available literature [29] or used field data [90]. Garcia et al. (2014) [91], and Ifaei et al. (2019) [92], and Safder et al. (2019) [93] chose a different approach including the balance of GHG in a wider frame of an environmental assessment by calculating the Ecoindicator 99 values and using it as one of the optimization criteria. Although very useful for the optimization scope in general, either approach leads to desired results if applied to particular multi-source and multi-fuel systems such as in the present study. Therefore, individual emission factors are best obtained by analyzing field data. An example of such an approach is the study by Jou et al. (2010) [82] who experimentally investigated the effect of changing fuel composition on both CO_2_ and NO_x_ emissions in a refinery furnace. Hadidi et al. (2016) [94] explored the mitigation options of GHG (CO_2_, NO_x_, SO_2,_ and volatile organic compounds) in a Saudi Arabian refinery with both help of mathematical modeling and input from refinery experts. Li et al. (2019) [95] studied the environmental performance of a high-temperature heat pump-assisted gas separation plant located in China within a more complex frame of energy, exergy, economic and environmental optimization of the given plant, yielding achievable savings in CO_2_, NO_x_, and SO_2_ emissions. Finally, the authors themselves applied the balances of CO_2_, NO_x,_ and SO_x_ emissions based on fuel composition and field emissions data in their earlier studies on refinery processes [38] or oxygen production plant [85] optimization. It can be concluded that, while all mentioned approaches are viable, using available field data ensures the validity of the environmental assessment results both generally and in a particular on-site application.

## 4. Conclusions

Steam crackers (ethylene plants) represent extraordinary material- and energy-integrated chemical complexes with significant potential for multiobjective optimization. Efforts aimed at cutting down the energy consumption and related GHG emissions focus mainly on SC´s steam network and often propose decreasing the steam let-down rates as well as reducing mechanical power requirement provided by the steam turbines on-site. Another option to optimize energy consumption is the export or import at multiple pressure levels if the SC is integrated into a chemical or refining complex. The presented study analyzed both the internal SC´s steam management as well as the steam import/export options for an SC integrated into a refinery. The exploitation of the first measure represents direct savings of fuel energy and is thus advantageous from the energetic, economic, and environmental points of view. Steam flows between the SC and the refinery result only from economically driven inter-fuel substitution between the SC steam boiler and those in the CHP of the refinery and their energetic and environmental effect is questionable. Thus, a detailed analysis of the change in the main air pollutants (CO_2_, NO_x_, SO_x_, CO) was performed to define the final observed changes in GHG emissions into changes attributable to individual measures.

A study carried out in a mid-capacity ethylene production plant integrated into a refinery revealed the following: 1. Optimization of the SC’s internal steam management by the installation of a smaller SS steam let-down valve minimized steam reduction rates and led to direct NG savings in the SC´s steam boiler. At 87 TJ/year, this reduction represents 5% of the annual steam boiler´s fuel consumption. The associated reduction of GHG emissions reached almost 4900 t/year and more than 5 t/year in the case of CO_2_ and NO_x_, respectively, while the reduction of both SO_x_ and CO emissions was negligible. The increase in the CO_2_ price affected the economics of this energy-saving measure beneficially. 2. Import of steam to the SC from the refinery due to lower fuel price used in the CHP boilers compared to that of NG, which led to an increase in the net consumption of fuel energy by 12 TJ/year. The reduction of CO_2_ emissions by more than 15,000 t/year and that of NO_x_ emissions by more than 15 t/year due to decreased fuel consumption in the SC boiler was outbalanced by the increase in GHG emissions in the CHP in the following annual amounts: CO_2_: more than 22,000 t; NO_x_ more than 30 t; SO_x_ more than 18 t. Thus, a net increase in emissions was observed for all investigated pollutants except for CO where the final effect was very close to zero. This means that the increase in the price of carbon emissions negatively affected the economics of steam import to the SC from the refinery. Therefore, the sole driver for steam import to the SC remained only the price differential between MPR and NG; in case of a reverse price situation, steam export from the SC is possible but it would be associated with difficulties in the HS steam management network.

The increase in the associated backpressure electricity production in the CHP due to the increase in steam import to the SC led to a decrease in external emissions, depending on the fuel mix considered. For that of Slovenské elektrárne, a.s., (Bratislava, Slovakia), the decrease in the external emissions was insufficient to counteract the increase in the net emissions in the SC and refinery complex. If, however, electricity production in a fossil-fuel power plant was considered, the external emissions decrease would outbalance the net emissions increase in the SC and refinery complex.

## Figures and Tables

**Figure 1 ijerph-18-12267-f001:**
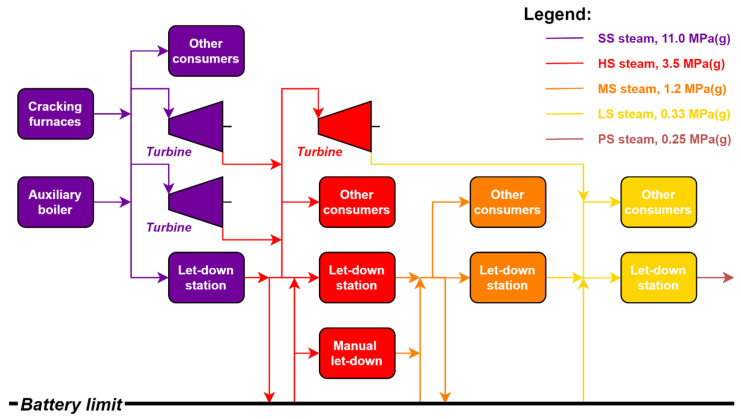
Simplified block scheme of the steam cracker´s (SC) steam network. HS—high-pressure steam, LS—low-pressure steam, MS—intermediate pressure steam, SS—superhigh pressure steam.

**Figure 2 ijerph-18-12267-f002:**
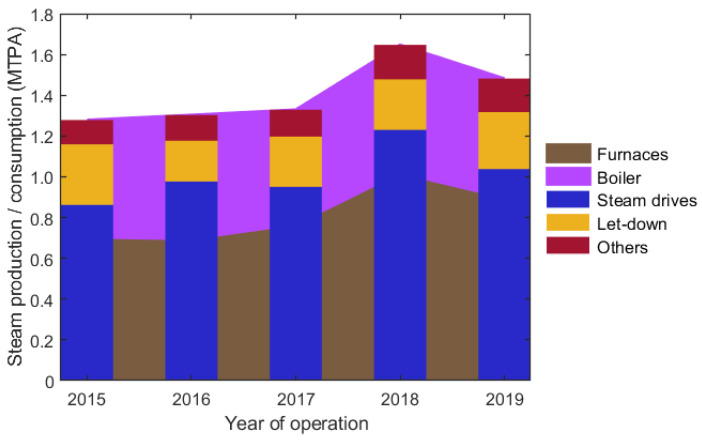
Annual SS steam production and consumption. MTPA—million tons per annum.

**Figure 3 ijerph-18-12267-f003:**
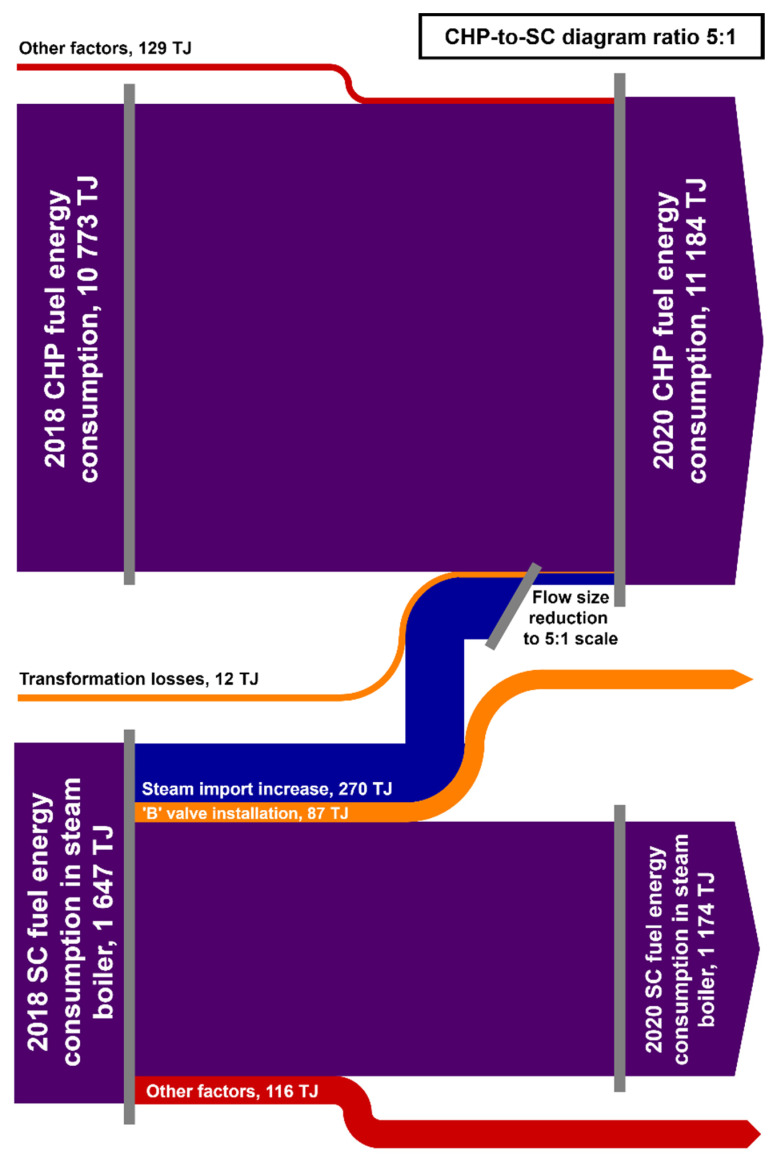
Comparison of fuel energy consumption in the SC’s steam boiler and the CHP’s (combined heat and power unit) steam boilers in 2018 and 2020.

**Figure 4 ijerph-18-12267-f004:**
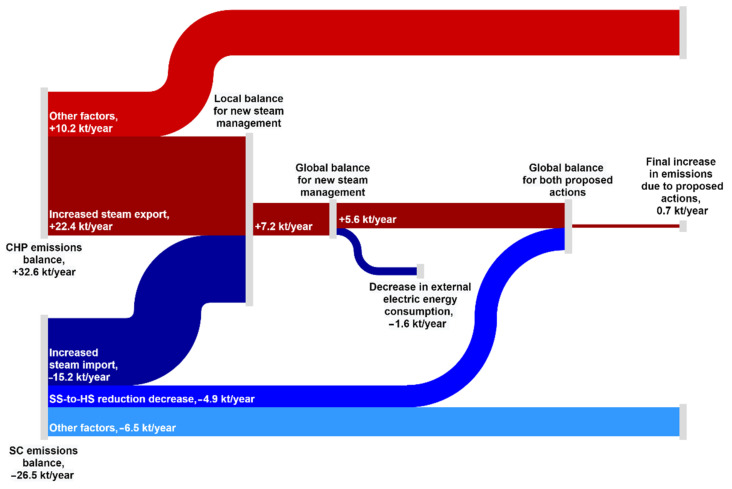
Effect of implemented energy management optimization on CO_2_ emissions balance change (2020 vs. 2018).

**Table 1 ijerph-18-12267-t001:** Annual SS steam production and consumption, and ethylene production. HS—high-pressure steam, SS—superhigh pressure steam.

Year	Specific Ethylene Production (% of Design Capacity)	SS Steam Consumption (t/% of Specific Ethylene Production)
2015	68	18,795
2016	73	17,849
2017	88	15,102
2018	100	16,460
2019	85	17,429

**Table 2 ijerph-18-12267-t002:** Steam let-down rate and valve opening before steam cracker (SC) turnaround in 2019; I to XII—January to December.

Time Period	Specific Ethylene Production (% of Quarterly Design Capacity)	SS Steam Let-Down to HS (t)	Let-Down Valve ‘A’ Opening (%)
2018 X to XII	94	55,264	15.10
2019 I to III	93	63,990	13.68
2019 IV to VI	106	78,924	14.19
2019 VII to IX	110	75,037	13.78
Quarterly average		68,304	

**Table 3 ijerph-18-12267-t003:** Key GHGs (greenhouse gases) emissions data with respect to steam cracker (SC) auxiliary steam boiler.

Fuel Energy Consumed in 2018	1647 TJ	
Pollutant	Emissions in 2018 (t)	Emission Factor (kg/TJ)
SO_x_	0.13	0.08
NO_x_	94.84	57.58
CO	1.787	1.09

**Table 4 ijerph-18-12267-t004:** Key GHGs emissions data on combined heat and power plant (CHP) steam boilers.

Fuel Energy Consumed in 2018	10,773 TJ	
Pollutant	Emissions in 2018 (t) ^1^	Emission Factor (kg/TJ)
SO_x_	705.55	65.50
NO_x_	1165.03	108.14
CO	10.28	0.95

^1^ t = tons.

**Table 5 ijerph-18-12267-t005:** Average emission factors of power produced by Slovenské Elektrárne, a.s., adapted from [38].

Pollutant	CO_2_	SO_x_	NO_x_	CO
Emission factor (kg/MWh)	136	0.392	0.107	0.061

**Table 6 ijerph-18-12267-t006:** Key process features and natural gas consumption decrease achieved by the installation and use of valve ‘B’.

Time Period	Specific Ethylene Production (% of Quarterly Design Capacity)	SS Steam Let-Down to HS (t) ^1^	SS to HS Steam Let-Down Valves Opening (%)		NG Consumption Decrease * (GJ)	NG Consumption Decrease * (t)
			Valve ‘A’	Valve ‘B’		
2020 I to III	108	47,326	1.58	69.51	8808	179.8
2020 IV to VI	106	15,136	0.01	46.31	28,198	575.5
2020 VII to IX	103	14,490	0.00	49.56	28,587	583.4
2020 X to XII	99	26,255	0.68	52.01	21,500	438.8
2020 total					87,094	1777.5

^1^ t = tons. * Compared to the average of 2018. NG—natural gas.

**Table 7 ijerph-18-12267-t007:** Effect of implemented energy management optimization on the balance of other GHGs (2020 vs. 2018).

Pollutant	CO	NO_x_	SO_x_
Effect of decreased SS to HS steam reduction, total (t) ^1^	−0.10	−5.02	−0.01
Effect of increased steam export to the SC, total (t), out of which	−0.02	14.95	18.45
Change in GHG emissions from the SC steam boiler (t)	−0.29	−15.54	−0.02
Change in GHG emissions from the CHP steam boilers (t)	0.27	30.49	18.47
Change in GHG emissions outside the refinery due to the increased steam export to the SC (t)	−0.71	−1.25	−4.57
Global change in GHG emissions due to the increased steam export to the SC (t)	−0.73	13.70	13.88

^1^ t = tons.

## Data Availability

All data obtained by calculations and analyses are listed directly in this study.

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
