# Peer review of "Energy and Environmental Assessment of Steam Management Optimization in an Ethylene Plant"

_ijerph, 2021, doi:10.3390/ijerph182212267_

Round 1

Reviewer 1 Report

The article "Energy and Environmental Assessment of Steam Management Optimization in an Ethylene Plant" presented by Miroslav Variny and co-authors is interesting and can be useful to the community studying greenhouse impacts and energy management. In my opinion, this article should be accepted after minor revision.

Authors started with a good introduction, with sufficient literature research.

Their methods are well described. They succeed to present their results.

However, there are some minor issues that require attention:

Page 1, Line 24:  ”ethylene plant + the refinery” what do you mean by “+”? it must be replaced by a suitable conjunction or word.

Page 1, Line 32: in the keywords, steam “S” in a small letter.

Lines 336, 337, 351, 389, 445, 518, 520: “+” should be replaced by an appropriate word.

Line 498: “5 %” with a space.

Author Response

We are very thankful for the reviewer´s suggestions and comments that helped to improve the quality of our manuscript. We endeavored to implement them properly. Major changes in the manuscript (modifications, newly added text) were highlighted in yellow. Detailed answers to individual reviewer´s comments are provided below.

Comments and Suggestions for Authors

The article "Energy and Environmental Assessment of Steam Management Optimization in an Ethylene Plant" presented by Miroslav Variny and co-authors is interesting and can be useful to the community studying greenhouse impacts and energy management. In my opinion, this article should be accepted after minor revision.

  1. Authors started with a good introduction, with sufficient literature research. Their methods are well described. They succeed to present their results.

Answer: Thank you very much for a positive opinion on our manuscript.

However, there are some minor issues that require attention:

  1. Page 1, Line 24:  ”ethylene plant + the refinery” what do you mean by “+”? it must be replaced by a suitable conjunction or word.

Answer: The “+” signs were replaced by “and” throughout the manuscript.

Page 1, Line 32: in the keywords, steam “S” in a small letter.

Answer: Given keyword was corrected.

Lines 336, 337, 351, 389, 445, 518, 520: “+” should be replaced by an appropriate word.

Answer: The “+” signs were replaced by “and” throughout the manuscript.

Line 498: “5 %” with a space.

Answer: Given expression was corrected.

Reviewer 2 Report

This article is very interesting and deserves to be published.

I simply regret that the authors have not further discussed about the price of carbon, its current volatility as well as the fact that, unless I am mistaken (but it will then be advisable to mention it for the reader anyway), the prices are today still too low to have much impact. 

Author Response

We are very thankful for the reviewer´s suggestions and comments that helped to improve the quality of our manuscript. We endeavored to implement them properly. Major changes in the manuscript (modifications, newly added text) were highlighted in yellow.

Comments and Suggestions for Authors

This article is very interesting and deserves to be published.

I simply regret that the authors have not further discussed about the price of carbon, its current volatility as well as the fact that, unless I am mistaken (but it will then be advisable to mention it for the reader anyway), the prices are today still too low to have much impact. 

Answer: Thank you very much for a truly positive opinion on our manuscript. Following your recommendation a few sentences touching this issue were added in part 3.3. of the revised manuscript.

Reviewer 3 Report

Dear Editor,

I had the pleasure of reading the manuscript entitled "Energy and Environmental Assessment of Steam Management Optimization in an Ethylene Plant". The introduction is clear and complete, the method is valid and the results are clearly exposed. For this reason I consider the article to be published as it is.

Author Response

Comments and Suggestions for Authors

Dear Editor,

I had the pleasure of reading the manuscript entitled "Energy and Environmental Assessment of Steam Management Optimization in an Ethylene Plant". The introduction is clear and complete, the method is valid and the results are clearly exposed. For this reason I consider the article to be published as it is.

Answer: Dear reviewer, thank you very much for an indeed positive opinion on our manuscript.

Reviewer 4 Report

This is a complete study, and all descriptions and analyses are correct and very well. Therefore, I think the current format can be published in IJERPH.

Author Response

We are very thankful for the reviewer´s suggestions and comments that helped to improve the quality of our manuscript. We endeavored to implement them properly. Major changes in the manuscript (modifications, newly added text) were highlighted in yellow.

English language and style

( ) Extensive editing of English language and style required
( ) Moderate English changes required
(x) English language and style are fine/minor spell check required
( ) I don't feel qualified to judge about the English language and style

Answer:  The manuscript was spell and grammar checked by a certified editor.

Comments and Suggestions for Authors

This is a complete study, and all descriptions and analyses are correct and very well. Therefore, I think the current format can be published in IJERPH.

 Answer: Dear reviewer, thank you very much for an indeed positive opinion on our manuscript.